# DEEP REINFORCEMENT LEARNING WITH IMPLICIT HUMAN FEEDBACK

## ABSTRACT

We consider the following central question in the field of Deep Reinforcement Learning (DRL): *How can we use implicit human feedback to accelerate and optimize the training of a DRL algorithm?* State-of-the-art methods rely on any human feedback to be provided explicitly, requiring the active participation of humans (e.g., expert labeling, demonstrations, etc.). In this work, we investigate an alternative paradigm, where non-expert humans are silently observing (and assessing) the agent interacting with the environment. The human's intrinsic reactions to the agent's behavior is sensed as implicit feedback by placing electrodes on the human scalp and monitoring what are known as event-related electric potentials. The implicit feedback is then used to augment the agent's learning in the RL tasks. We develop a system to obtain and accurately decode the implicit human feedback (specifically error-related event potentials) for state-action pairs in an Atari-type environment. As a baseline contribution, we demonstrate the feasibility of capturing error-potentials of a human observer watching an agent learning to play several different Atari-games using an electroencephalogram (EEG) cap, and then decoding the signals appropriately and using them as an auxiliary reward function to a DRL algorithm with the intent of accelerating its learning of the game. Building atop the baseline, we then make the following novel contributions in our work: (i) We argue that the definition of error-potentials is *generalizable* across different environments; specifically we show that error-potentials of an observer can be learned for a specific game, and the definition used as-is for another game without requiring re-learning of the error-potentials. (ii) We propose two different frameworks to combine recent advances in DRL into the error-potential based feedback system in a sample-efficient manner, allowing humans to provide implicit feedback while training in the loop, or prior to the training of the RL agent. (iii) Finally, we scale the implicit human feedback (via ErrP) based RL to reasonably complex environments (games) and demonstrate the significance of our approach through synthetic and real user experiments.

## 1 INTRODUCTION

Deep Reinforcement Learning (DRL) algorithms have now beaten human experts in Go (Silver et al., 2017), taught robots to become parkour masters (Heess et al., 2017), and enabled truly autonomous vehicles (Wang et al., 2018). However, current state-of-the-art RL agents equipped with deep neural networks are inherently complex, difficult and time-intensive to train. Particularly in complex environments with sparse reward functions (e.g., maze navigation), the DRL agents need an inordinate amount of interaction with the environment to learn the optimal policy. Human participation can potentially help DRL algorithms by accelerating their training and reducing the learning costs without compromising final performance. This potential has inspired a several research efforts where either an alternative (or supplementary) feedback is obtained from the human participant (Knox, 2012). Such approaches despite being highly effective, severely burden the human-in-the-loop demanding either expert demonstrations (Ross et al., 2011) or explicit feedback (Christiano et al., 2017).

In this paper, we investigate an alternative paradigm that substantially increases the richness of the reward functions, while not severely burdening the human-in-the-loop. We study the use of electroencephalogram (EEG) based brain waves of the human-in-the-loop to generate the reward functions that can be used by the DRL algorithms. Such a model will benefit from the natural rich activity

of a powerful sensor (the human brain), but at the same time not burden the human if the activity being relied upon is *intrinsic*. This paradigm is inspired by a high-level error-processing system in humans that generates error-related potential/negativity (ErrP or ERN) (Scheffers et al., 1996).When a human recognizes an error made by an agent, the elicited ErrP can be captured through EEG to inform agent about the sub-optimality of the taken action in the particular state.

As a baseline contribution, we demonstrate the feasibility of capturing error-potentials of a human observer watching an agent learning to play several different Atari-games, and then decoding the signals appropriately and using them as an auxiliary reward function to a DRL algorithm. We show that a *full access* approach to obtain feedback on every state-action pair while RL agent is learning, can significantly speedup the training convergence of RL agent. We contend that while obtaining such implicit human feedback through EEG is less burdensome, it is still a time-intensive task for the subject and the experimenter alike. This, combined with the noisy EEG signals and stochasticity in inferring error-potentials, raises significant challenges in terms of the practicality of the solution.

In this context, we first argue that the definition of ErrPs is generalizable across different environments. We show that ErrPs of an observer can be learned for a specific game, and the definition used as-is for another game without requiring re-learning of the ErrP. This is notably different from previous approaches (Chavarriaga & Millán, 2010; Salazar-Gomez et al., 2017), where the labeled ErrPs are obtained in the same environment (where the RL task is performed). For any new and unseen environment, it does not require the human to go through the training phase again, and assumes no prior knowledge about the optimal state-action pairs of the environment.

We present two different frameworks to combine recent advances in DRL into the implicit human feedback mechanism (via ErrP) in a practical, sample-efficient manner. This reduces the cost of human supervision sufficiently allowing the DRL systems to train. Relying on Active Learning (AL) methods, our first framework allows humans to provide implicit feedback in the loop, while an RL agent is being trained. An uncertainty based acquisition function is modeled to select the samples state-action pairs for querying the implicit human feedback. However, as a human is always required to be in the loop, our second framework allows humans to provide their feedback implicitly before the agent starts training. Based on the human feedback obtained during pre-training, a quality (Q) function is learned over these imperfect demonstrations to provide the supplementary reward to the RL agent. We present results from real ErrP experiments to evaluate the acceleration in learning, and sample efficiency, in both frameworks. In summary, the novel contributions of our work are,

1. We demonstrate the generalizability of error-potentials over various Atari-like environments (discrete grid-based navigation games, studied in this work), enabling the estimation of implicit human feedback in new and unseen environments.
2. We propose two different frameworks to combine recent advances in DRL into ErrP based feedback system in a practical, sample-efficient manner. The first framework allows humans to provide implicit feedback while training in the loop. Taking advantage of recent approaches in learning from imperfect demonstrations, in the second framework, the implicit human feedback is obtained prior to the training of the RL agent.
3. We scale the implicit human feedback (via ErrP) based RL to reasonably complex environments and demonstrate the significance of our approach through synthetic and real user experiments.

## 1.1 RELATED WORK

Daniel et al. (2015); El Asri et al. (2016); Wang et al. (2016) studied RL from human rankings or ratings, however rely on explicit human feedback, and assume that the feedback is noiseless. Demonstrations have been commonly used to improve the efficiency of RL (Kim et al., 2013; Chemali & Lazaric, 2015; Piot et al., 2014), and a common paradigm is to initialize RL algorithms with good policy or Q function (Nair et al., 2018; Hester et al., 2018; Gao et al., 2018). In this work, we use rely on implicit feedback from non-expert humans (via ErrPs) which is inherently noisy.

(Chavarriaga & Millán, 2010; Iturrate et al., 2010; Salazar-Gomez et al., 2017) demonstrate the benefit of ErrPs in a very simple setting (i.e., very small state-space), and use ErrP-based feedback as the only reward. Moreover, in all of these works, the ErrP decoder is trained on a similar game (or robotic task), essentially using the knowledge that is supposed to be unknown in the RL task. In our work, we use labeled ErrPs examples of very simple and known environments to train the ErrP decoder, and combine with the recent advances in DRL in a sample-efficient manner for reasonably complex environments.

Figure 1: Manifestation of error-potentials in time-domain: Grand average potentials (error-minus-correct conditions) are shown for Maze, Catch and Wobble game environments. Thick black line denotes the average over all the subjects. The game environments are explained in section

## 2 DEFINITIONS AND PRELIMINARIES

Consider a Markov Decision Process (MDP) problem $M$, as a tuple $< \mathcal{X}, \mathcal{A}, P, P_0, R, \gamma >$, with state-space $\mathcal{X}$, action-space $\mathcal{A}$, transition kernel $P$, initial state distribution $P_0$, accompanied with reward function $R$, and discounting factor $0 \leq \gamma \leq 1$. Here the random variable $Z(\boldsymbol{s}, \boldsymbol{a})$ denotes the accumulated discounted future rewards starting from state $\boldsymbol{s}$ and action $\boldsymbol{a}$.

In this work, we only consider MDP with discrete actions and states. In model-free RL method, the central idea of most prominent approaches is to learn the Q-function by minimizing the Bellman residual, i.e., $\mathcal{L}(Q) = \mathbb{E}_{\pi}\big[\big(Q(x, a) - r - \gamma Q(x', \hat{a})\big)^2\big]$, and temporal difference (TD) (Tesauro, 1995) update where the transition tuple $(x, a, r, x')$ consists of a consecutive experience under behavior policy $\pi$. Modern techniques in DRL such as DQN (Mnih et al., 2015) and the target network (Van Hasselt et al., 2016) are also adpoted throughout the paper.

## 3 INTEGRATING DRL WITH IMPLICIT HUMAN FEEDBACK: AN IDEAL APPROACH

The humans intrinsic reactions to the agents behavior is sensed as implicit feedback by placing electrodes on the human scalp and monitoring what are known as event-related electric potentials. We rely on the Riemannian Geometry framework for the classification of error-related potentials (Barachant & Congedo, 2014; Congedo et al., 2013) presented in Appendix 7.1. We consider the classification of error-related potentials as a binary variable indicating the presence (i.e., action taken by the agent is incorrect) and absence of error (i.e., action taken by the agent is correct).

With the availability of implicit human feedback, we explore how the training of state-of-the-art DRL algorithms can be accelerated. A trivial approach is to obtain feedback on every state-action pair while RL agent is learning (also known as *full access*). It is to add a negative penalty to the reward when ErrP is detected, and keep using the original reward from the environment without ErrP detected. The evaluation result of this method based on real ErrP data is shown in section 5.1. The results validate that this method can speed up the training convergence of RL agent significantly.

We contend that while obtaining such implicit human feedback through EEG is less burdensome, it is still a time-intensive task for the subject and the experimenter alike. This, combined with the noisy EEG signals and stochasticity in inferring ErrPs, raises significant challenges in terms of the practicality of the solution.

## 4 TOWARDS PRACTICAL INTEGRATION OF DRL WITH IMPLICIT HUMAN FEEDBACK

In this section, we discuss three approaches towards integrating the ErrP with recent advances in DRL in a practical manner. Firstly, we show that ErrPs of an observer can be learned for a specific game, and the definition used as-is for another game without requiring re-learning of the ErrP. Further, we discuss two frameworks to combine the recent advances in DRL into the implicit human feedback mechanism (via ErrP) to accelerate the RL agent learning in a sample-efficient manner. The first framework allows humans to provide implicit feedback while training in the loop, without any prior knowledge on the game. In the second framework, the implicit human feedback is obtained prior to the training of the RL agent. It exploits the initially given trajectories with ErrP labels to learn a reward function for augmenting the RL agent, where human with some prior knowledge is needed to specify some non-expert trajectories. Recently, Q function can be shown to have better generalization in state-space if trained with non-expert demonstrations (Luo et al., 2019).

## 4.1 ErrP Generalization across Environments

Error-potentials in the EEG signals is studied under two major paradigms in human-machine interaction tasks, (i) *feedback and response ErrPs:* error made by human (Carter et al., 1998; Falkenstein et al., 2000; Blankertz et al., 2003; Parra et al., 2003; Holroyd & Coles, 2002), (ii) *interaction ErrPs:* error made by machine in interpreting human intent (Ferrez & Millán, 2005). Another interesting paradigm is when human is watching (and silently assessing) the machine performing a specific task (Chavarriaga & Millán, 2010). The manifestation of these potentials across these paradigms were found quite similar in terms of their general shape, timings of negative and positive peaks, frequency characteristics etc., (Ferrez & Millán, 2005; Chavarriaga & Millán, 2010). This prompts us to explore the consistency of the error-potentials across different environments (i.e., games, in our case). We restrict the score of our work to the paradigm of human acting as a silent observer of the machine actions. In Fig.5, we plot the grand average waveforms across three environments (Maze, Catch and Wobble), to visually validate the consistency of potentials. We can see that the shape of negativity, and the timings of the peaks is quite consistent across the three game environments studied in this work. Further, we perform experimental evaluation in section 5.2.1, to show that error-potentials are indeed generalizable across environments, and can further be used to inform deep reinforcement learning algorithm in a new and unseen environments.

## 4.2 First framework: Training with Implicit Human Feedback in the loop

Active Learning (AL) frameworks have been proved quite successful in optimizing the learning task while minimizing the required number of labeled examples (Cohn et al., 1996; Gal et al., 2017). In AL, an acquisition function is used to efficiently select the data points requested for labeling from an external oracle. We introduce a framework of training RL agents with implicit non-expert human feedback in the loop, leveraging recent advances in active learning methods.

We present our active learning based framework in Fig. 2(a). We use an uncertainty-based acquisition function to select the state-action pairs required for non-expert human labeling (via ErrP). Since it is critical to keep the coherence between consecutive state-action pairs shown to the human subject, a full trajectory from start to end of the game can be shown. The calculation of the acquisition function is based on the state-action pair uncertainty along the trajectory, as explained in Appendix 7.3. Specifically, we model the Deep-Q-Network (DQN) by Bayesian learning methods, which have strong capacity of uncertainty estimation (Gal et al., 2017). The DQN is trained with experience collected in the reply buffer, a structure commonly used in deep RL algorithms.

In contrast to the *full access* method, the presented framework queries for ErrP based state-action pair labeling only at the end of every $N_E$ episodes. We further store the decoded ErrP labels into buckets, to be used for future training augmentation. In every step, the RL agent inquire the negativity of the current state-action pair from buckets, instead of ErrP labeling, which reduces the number of ErrP inquiries significantly. This negativity can add a negative penalty to the environmental reward as auxiliary.

**Trajectory Generation and Selection:** ErrP labeling informs the RL agent about negativity of selected actions, ideally preventing the agent from deviating from the optimal paths in the game. However, these optimal paths are unknown a priori. For generating trajectories for ErrP labeling, we empirically found that following greedily the action with largest Q value in every state based on the most updated DQN performs very well. Then the trajectory with the largest acquisition function output is selected for querying ErrP labels. Three acquisition functions evaluated in experiments are all formulated based on the uncertainty estimation of Q values, and their formulations and approximations are introduced in Appendix 7.3. The framework are presented in Algorithm 1.

## 4.3 Second Framework: Learning from Imperfect Demonstrations with Implicit Human Feedback

RL algorithms deployed in the environment with sparse rewards demand heavy explorations (require a large number of trial-and-errors) during the initial stages of training. Imitation learning from a small number of demonstrations followed by RL fine-tuning is a promising paradigm to improve the sample efficiency in such cases (Večerík et al., 2017; Hester et al., 2018; Gao et al., 2018). Inspired by the paradigm of imitation learning, we develop a novel framework that can robustly

---

**Algorithm 1:** Integrating Implicit Human Feedback while Training

---

**Input:** Parameters $N_E, N_T$, acquisition function $a(\cdot)$

1 Initialize DQN $\mathcal{Q}(\cdot, \cdot)$ ;
2 **for** *episode=1,2,...* **do**
3     Starting from random initial state, the RL agent plays the game until the end of the episode;
4     Update the DQN $\mathcal{Q}$ by experiences randomly selected from the replay buffer ;
5     In every $N_E$ episode: ;
6         Generate $N_T$ trajectories $\{\tau_k\}_{k=1}^{N_T}$ by following the actions induced from $\mathcal{Q}$ ;
7         Select the trajectory maximizing the acquisition function $a(\cdot)$ for obtaining ErrP data;
8         Return the decoded ErrP labels to the RL agent for future training augmentation ;
9 **end**

---

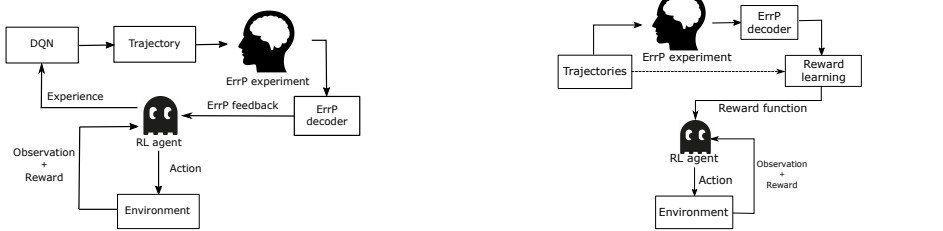

(a) Training with implicit human feedback in the loop. Trajectory block: trajectory generation and selection

(b) Learning from imperfect demonstration with implicit human feedback. The dashed line shows trajectories in $\mathcal{D} \cup \mathcal{D}_R$, and are used in reward learning

Figure 2: Integrating DRL with Implicit Human Feedback

learn a reward function to augment the DRL algorithms and accelerate the training of RL agent. This reward function is derived from reward function with imperfect demonstrations, achieved by obtaining the implicit human feedback in the form of ErrP over a set of trajectories.

The flowchart of the second framework is in Fig. 2(b). In this framework, the trajectories in the demonstration are first criticized by ErrP labeling in experiments, and a quality (Q) function is learned from the labeled trajectories in the reward learning step. An alternative reward is derived from the learned quality function, augmenting the following RL algorithm. This approach is considerably different from our first framework (section 4.2), as we only make queries for ErrP labeling on trajectories initially given in the demonstration (rather than making queries continuously during every training step). These queries are made before the RL agent starts training, improving the efficiency of the total number of labeling (implicit, ErrP based) queries made to the external oracle (human). Similar to the first framework, the demonstrations for ErrP labeling can only consist of complete trajectories. We assumed that the trajectories in the demonstration are initially specified by human or other external sources, without any reward information. This is a reasonable assumption since the rewards may be unknown to humans in general cases. The human subject in the experiment provides feedback in an implicit manner (via ErrP) on state-action pairs along the trajectories, labeling every state-action pair as a *positive* or *negative* sample. Based on the decoded ErrP labels and initially given trajectories, the proposed framework learns the reward function based on maximum entropy RL methods (Ziebart, 2010), as explained in details in Appendix 7.4.

Different from conventional imitation learning, these trajectories are not given by expert policies, allowing the non-experts to demonstrate. Moreover, the Q function learned from imperfect demonstrations can have better estimations on states unseen in the demonstration, and provide better generalization in the state-space (Luo et al., 2019).

## 5 EVALUATION

We have developed three discrete-grid based navigation games in OpenAI Gym emulating *Atari* framework (Brockman et al., 2016), namely (i) Wobble, (ii) Catch, and (iii) Maze, shown in Fig. 3(a). We use the default Atari dimensions (i.e., 210x160 pixels). The source codes of the games can be found in the public repository[1], and can be used with the OpenAI Gym module.

---

[1] source code is attached with the submission for anonymity purposes

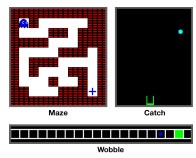 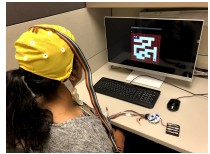 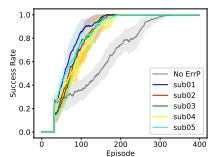 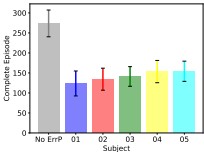

(a) Game Environments    (b) Experiment Bench          (a) Learning Curve    (b) Complete Episode

Figure 3: Experimental framework          Figure 4: RL with *full access* to ErrP feedback.

**Wobble:** Wobble is a simple 1-D cursor-target game, where the middle horizontal plane is divided into 20 discrete blocks. At the beginning of the game, the cursor appears at the center of the screen, and the target appears no more than three blocks away from the cursor position. The action space for the agent is moving one step either left or right. The game is finished when the cursor reaches the target. Once the game is finished, a new game is started with the cursor in place.

**Catch:** Catch is a simplistic version of *Eggomania*[2] (Atari 2600 benchmark), where we display a single egg on the screen at a time. The screen dimensions are divided into 10x10 grid space, where the *egg* and the *cart*, both occupies one block. The action space of the agent consists of "NOOP" (no operation), "moving left" and "moving right". At the start of the game, the horizontal position of the egg is chosen randomly. At each time step, the *egg* falls one block in the vertical direction.

**Maze:** Maze is a 2-D navigational game, where the agent has to reach to a fixed target. The Atari screen is centered and divided into 10x10 equal-sized blocks. The agent and target occupy one block. The action space consists of four directional movements. The maze architecture is kept fixed for the purpose of this work. If an agent moves, but hits a wall, a quick blinking of the agent is displayed, to show the action taken by the agent.

**EEG experimental protocol:** We designed and developed an experimental protocol, where a machine agent plays a computer game, while a human silently observes (and assesses) the actions taken by the machine agent. These implicit human reactions are captured by placing raw electrodes on the scalp of the human brain in the form of EEG. The electrode cap was attached with the OpenBCI[3] platform, which was further connected to a desktop machine over the wireless channel. In the game design (developed on OpenAI Gym), we open a TCP port, and continuously transmit the current state-action pair using the TCP/IP protocol. We used OpenViBE software (Renard et al., 2010) to record the human EEG data. OpenViBE continuously listens to the TCP port (for state-action pairs), and timestamps the EEG data in a synchronized manner. A total of five human subjects were recruited using standard procedures. We recruited five human subjects (mean age $26.8 \pm 1.92$, 1 female) for collecting the EEG data. For each subject, we conducted three separate sessions over multiple days. For each subject-game pair, the experimental duration was less than 15 minutes. The agent took action every 1.5 seconds. All the research protocols for the user data collection were reviewed and approved by the Institutional Review Board[4].

## 5.1 FULL ACCESS

The *full access* method as discussed in section 3 is the most preliminary approach to make ErrP labels augment the RL algorithm. It has the fastest training convergence rate (provides upper bound) but makes the maximum possible queries to the external oracle (human) for the implicit feedback. We use this method as a benchmark for comparing the data-efficiency of other RL augmentation methods. The results with real ErrP data of 5 subjects are shown in Figure 4. Here the training data of ErrP decoder is from Catch game while the testing data is from Maze. We can see there is a significant improvement in the training convergence. It further validates the generalization capability of ErrP decoding from 1-D to 2-D navigation games. In this paper, "No ErrP" method refers to regular RL algorithms without the help of any human feedback. The success rate is defined as the ratio of success plays in the previous 32 episodes. The training completes when the success rate reaches to 1. In all plots of this paper, solid lines are average values over 10 random seeds, and shaded regions correspond to one standard deviation. In the evaluations of this paper, the Q network is modeled by Bayesian deep learning methods, such as Bayesian DQN or bootstrapped DQN, introduced in Appendix 7.2.

---

[2]https://en.wikipedia.org/wiki/Eggomania

[3]http://openbci.com

[4]The Institution name is not disclosed to ensure the anonymity of the author affiliations.

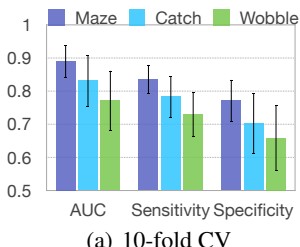
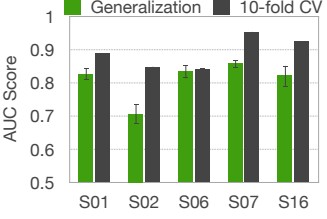
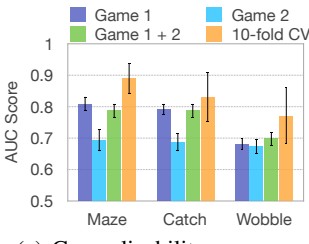

(a) 10-fold CV    (b) Generalizability over subjects    (c) Generalizability over games

Figure 5: Detection performance and generalizability of ErrP: (a) 10-fold CV performance of each game i.e., no generalization, (b) generalizability from Catch to Maze over subjects compared with 10-fold CV, (c) generalizability over all combinations of three games compared with 10-fold CV.

## 5.2 PRACTICAL SOLUTION

In this subsection, we evaluate the performance of three approaches to practially integrate the DRL with implicit human feedback (via ErrPs).

### 5.2.1 GENERALIZABILITY

We first validate the feasibility of decoding ErrP signals using a 10-fold cross-validation scheme for each game. In this scheme, we train and test on the ErrP samples of the same game environment. In Fig. 5(a), we show the performance of three games in terms of AUC score, sensitivity and specificity, averaged over 5 subjects. The Maze game has the highest AUC score ($0.89 \pm 0.05$) followed by Catch ($0.83 \pm 0.08$) and Wobble ($0.77 \pm 0.09$). To evaluate the generalization capability of error-potential signals and the decoding algorithm, we train on the samples collected from Catch and test on Maze game. In Fig. 5(b), we provide the AUC score performance compared with the 10-fold CV AUC score of Maze. We can see that the Catch game is able to capture more than 80% of the variability in the ErrPs for Maze game. To provide deeper insights into the generalizability extent, we present the AUC score of generalizability performance over all combinations in fig. 5(c). In the later subsections, we experimentally show that these performance numbers are sufficient to achieve 2.25x improvement in training time (in terms of the number of episodes required).

We performed preliminary experiments to gain fundamental insights into the extent of generalizability. All the three games considered in this work, differ in terms of their action space. Wobble can move either left or right (two actions), Catch has an additional "NOOP" (3 actions), and the agent in the Maze can move in either direction (4 actions). To understand the generalizability of ErrP in terms of the actions taken by the agent, we train on the Wobble, and test on the Catch game for two groups - (i) when the agent moves in either direction, and (ii) when the agent stays in the place. We obtain an average AUC score of $0.7359$ ($\pm 0.1294$) and $0.6423$ ($\pm 0.1451$) for both groups, respectively. Through a paired t-test, we found the difference in mean statistically significant. Similarly, for the Catch game, we test two groups - (i) when *egg* is close to the *paddle*, and (ii) when *egg* is far from the *paddle*. We found the mean AUC scores of $0.71$ ($\pm 0.1$) and $0.84$ ($\pm 0.12$) for each group, respectively. The difference of the mean of both groups was found statistically significant.

### 5.2.2 EVALUATION OF FIRST FRAMEWORK

In evaluating active RL framework, we explore three forms of acquisition functions, i.e., entropy, mutual information, and confidence interval. Their expressions and approximation techniques are illustrated with details in Appendix 7.3. The benchmark performance of *full access* method is shown in section 5.1. We first evaluate the performance of first framework with synthesized human feedback, which is presented in Appendix 7.5.1 on box world environment(Zambaldi et al., 2018).

In this section, we evaluate the first framework on Maze game with real ErrP experimental data. We use Bayesian DQN for the Q network. Three acquisition functions are compared in Figure 6 with detailed statistics on Table 7.5.1, which has similar conclusions as the synthetic case. Based on real ErrP data, we can show that compared with full access method, the first framework can reach similar performance with much less feedback inquiries.

### 5.2.3 EVALUATION OF SECOND FRAMEWORK

In the evaluation of this framework, the trajectories given initially are generated based on optimal paths randomly corrupted by wrong actions, which appear with the probability of 0.2. We evaluate

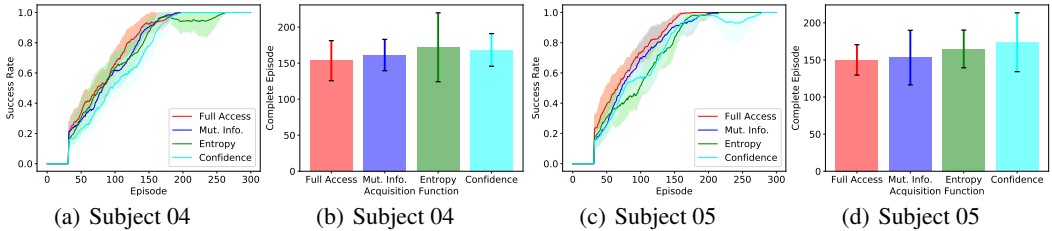

(a) Subject 04      (b) Subject 04      (c) Subject 05      (d) Subject 05

Figure 6: Evaluation of First Framework: Training with Implicit Human Feedback in the loop. Comparison of Three Acquisition Functions in Figure (b) and (d).

the performance with 10 and 20 trajectories given initially. Prior to training the RL agent, each subject is asked to provide feedback via ErrP on the state-action pairs along these trajectories. We conducted experiments on 5 subjects, based on Maze game. Here the Q network is modeled by Bayesian DQN. The performance of augmented RL algorithms is shown in Figure 7.

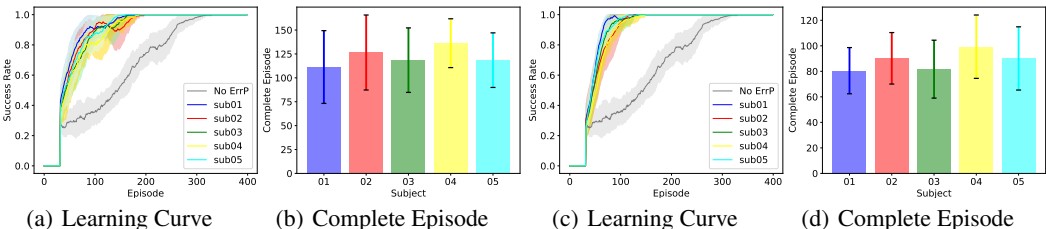

(a) Learning Curve      (b) Complete Episode      (c) Learning Curve      (d) Complete Episode

Figure 7: Evaluation of Second Framework: Learning from Imperfect Demonstrations Labeled by ErrP. Figures (a) and (b) are for the demonstration with 10 trajectories, and figures (c) and (d) are for 20 trajectories.

The reward function is shown to speed up the training convergence of the RL agent significantly. Since trajectories are randomly generated initially, the number of ErrP inquiries of the second framework is equal to $372.1(\pm 58.2)$, based on the statistics in our simulations. The second framework even outperforms the full access method, with ErrP inquiries on only 20 trajectories, proving its data efficiency. However, this framework needs a human or external source, who has some prior knowledge of the game, to specify the initial trajectories.

## 6 CONCLUSIONS AND FUTURE WORK

We first demonstrate the feasibility of capturing error-potentials of a human observer watching an agent learning to play several different Atari-games, and then decoding the signals appropriately and using them as an auxiliary reward function to a DRL algorithm. Then we argue that the definition of ErrPs is generalizable across different environment. In the ideal approach, we validate the augmentation effect of ErrP labels on RL algorithms by the *full access* method. Then, in the practical approach, we propose two augmentation frameworks for RL agent, applicable to different situations. The first is to integrate human into the training loop of RL agent based on active learning, while the second is to learn a reward function from imperfect demonstrations labeled by ErrP.

The demonstration of the generalizability of error-potentials is limited across the environments presented in the paper. We have considered discrete grid-based reasonably complex navigation games. The validation of the generalization to a variety of Atari and Robotic environments is the subject of the future work. We also plan to test our framework of integrating implicit human feedback (via ErrPs) over robotic environments, and text the *generalization* capability of error-potentials between virtual and physical worlds.

As future work, we plan to investigate as to how machines can be assisted in RL by using intrinsic EEG-based cooperations among humans and machines.

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

# 7 APPENDIX

## 7.1 OBTAINING THE IMPLICIT HUMAN FEEDBACK: DECODING ERRPS

The Riemannian Geometry based framework was first proposed in (Barachant & Congedo, 2014; Congedo et al., 2013) to translate teh raw EEG signals into meaningful labels[5]. The raw EEG data

---

[5]The authors successfully applied the framework and won multiple Kaggle challenges. E.g., https://www.kaggle.com/c/inria-bci-challenge. Later, this framework was successfully adapted in many other error-potential decoding works (Salazar-Gomez et al., 2017).

are bandpass filtered in [0.5, 40] Hz. Epochs of 800ms were extracted relative to pre-stimulus 200ms baseline, and were subjected to spatial filtering. In spatial filtering, prototype responses of each class, i.e., "correct" and "erroneous", are computed by averaging all training trials in the corresponding classes("xDAWN Spatial Filter" (Rivet et al., 2009; Barachant & Congedo, 2014; Congedo et al., 2013)). "xDAWN filtering" projects the EEG signals from sensor space (i.e., electrode space) to the source space (i.e., a low-dimensional space constituted by the actual neuronal ensembles in brain firing coherently). The covariance matrix of each epoch is computed, and concatenated with the prototype responses of the class. Further, dimensionality reduction is achieved by selecting relevant channels through backward elimination (Barachant & Bonnet, 2011). The filtered signals are projected to the tangent space (Barachant et al., 2013; 2011) for feature extraction. The obtained feature vector is first normalized (using L1 norm) and fed to a regularized regression model. A threshold value is selected for the final decision by maximizing accuracy offline on the training set. We present the algorithm to decode the ErrP signals in Algorithm 2.

---

**Algorithm 2:** Riemannian Geometry based ErrP classification algorithm (Barachant et al., 2013)

**Input** : raw EEG signals $EEG$

1 Pre-process raw EEG signals ;
2 Spatial Filtering: xDAWN Spatial Filter ($nfilter$) ;
3 Electrode Selection: ElectrodeSelect ($nelec$, metric='riemann') ;
4 Tangent Space Projection : TangentSpace(metric = "logeuclid") Normalize using L1 norm ;
5 Regression: ElasticNet ;
6 Select decision threshold by maximizing accuracy

---

### 7.2 Deep Q Network Models

Here we introduce two DQN models adopted in this paper.

**Bayesian DQN** The first model we use is a DQN architecture where the Q-function is approximated as a linear function, with weights $\omega_a$, of the feature representation of states $\phi_\theta(x) \in \mathbb{R}^d$, parameterized by neural network with weights $\theta$ (Osband et al., 2013). Here by utilizing the DQN architecture and imposing Gaussian distributions on $\omega_a$, based on Bayesian linear regression (BLR) (Rasmussen, 2003), the posterior of $\omega_a$ can be calculated by

$$\omega_a \sim \mathcal{N}(\bar{\omega}_a, \mathrm{Cov}_a), \quad \bar{\omega}_a := \frac{1}{\sigma_\epsilon^2}\mathrm{Cov}_a\Phi_a^\theta \boldsymbol{y}_a, \quad \mathrm{Cov}_a := \left(\frac{1}{\sigma_\epsilon^2}\Phi_a^\theta\Phi_a^{\theta T} + \frac{1}{\sigma^2}I\right)^{-1} \tag{1}$$

where we construct disjoint replay buffer $\mathcal{D}_a$ corresponding to experience with action $a$, and a matrix $\Phi_a^\theta \in \mathbb{R}^{d\times|\mathcal{D}_a|}$, vector $\boldsymbol{y}_a$, i.e., the concatenation of state features and target values in set $\mathcal{D}_a$. Therefore the posterior of Q value can be the following the Gaussian distribution,

$$Q(x, a) \sim \mathcal{N}(\bar{\omega}_a^T\phi_\theta(x), \phi_\theta(x)^T\mathrm{Cov}_a\phi_\theta(x)) \tag{2}$$

**Bootstrapped DQN** Another Bayesian DQN model we use is bootstrapped DQN (Osband et al., 2016). It explores in a similar manner as the Bayesian DQN introduced above, but uses a bootstrapped neural network to approximate a posterior sample for the value. Bootstrapped DQN is also provably efficient, but adopts neural network instead of linear value function and bootstraps instead of Gaussian sampling. It is implemented by $K \in \mathbb{N}$ bootstrapped estimates of the Q value in parallel, i.e., $Q_k(s, a; \theta), s = 1, \ldots, K$.

### 7.3 Acquisition Function

In this work, three acquisition functions are explored in selecting trajectory for ErrP labeling. The trajectory is defined as a sequence of state-action pairs $\tau := \{(\boldsymbol{s}_0, \boldsymbol{a}_0), \ldots, (\boldsymbol{s}_T, \boldsymbol{a}_T)\}$. We denote the trajectory set as $\mathcal{D}$, the learned Q network as $\mathcal{Q}$, and acquisition function as $a(\tau, \mathcal{Q})$. The selected trajectory maximizes the acquisition function given $\mathcal{Q}$. In this work, we explore three acquisition functions:

- *Entropy*: Select the trajectory with the maximum entropy, which measures the uncertainty of state-action pairs along the trajectory.

$$a(\tau, \mathcal{Q}) := \mathbb{H}[\tau|\mathcal{Q}] = -\sum_{t=0}^{T} \mathbb{H}[\boldsymbol{s}_t, \boldsymbol{a}_t|\mathcal{Q}]$$

  Specifically, in Bayesian and bootstrapped DQN, since Q value is approximately Gaussian distributed, we can directly use the differential entropy of Gaussian random variable here, i.e., $\mathbb{H}[\boldsymbol{s}_t, \boldsymbol{a}_t|\mathcal{Q}] = \frac{1}{2}\log 2\pi e \mathrm{Cov}_a$, where the variance term $\mathrm{Cov}_a$ can be replaced by equation 1 or the variance of $K$ heads of bootstrapped DQN.

- *Mutual Information*: choose the trajectory with maximum mutual information between priors and posterior of state-action distributions (Houlsby et al., 2011). The mutual information represents the information gain of action selections. We have

$$a(\tau, \mathcal{Q}) := \mathbb{I}[\tau|\mathcal{Q}] = \sum_{t=0}^{T} \mathbb{I}[\boldsymbol{s}_t, \boldsymbol{a}_t|\mathcal{Q}]$$

  Following (Nikolov et al., 2018), we approximate the mutual information $\mathbb{I}[\boldsymbol{s}_t, \boldsymbol{a}_t|\mathcal{Q}]$ by $\log(1+\frac{\sigma(\boldsymbol{s}_t,\boldsymbol{a}_t)^2}{\rho(\boldsymbol{s}_t,\boldsymbol{a}_t)^2})$, where $\sigma(\boldsymbol{s}_t, \boldsymbol{a}_t)$ characterizes the *parametric uncertainty* of Q value, and $\rho(\boldsymbol{s}_t, \boldsymbol{a}_t)^2$ represents the *intrinsic uncertainty*. In this work, $\sigma(\cdot, \cdot)^2$ is obtained by $\mathrm{Cov}_a$ in equation 1 for Bayesian DQN, and by the variance of $K$ heads in bootstrapped DQN. We approximate $\rho(\boldsymbol{s}_t, \boldsymbol{a}_t)$ by the variance of accumulated rewards $Z(\boldsymbol{s}_t, \boldsymbol{a}_t)$, obtained by histogram statistics, C51 (Bellemare et al., 2017) or QR-DQN (Dabney et al., 2018).

- *Confidence Interval*: Choose the trajectory with the maximum uncertainty in ErrP labeling. Some state-action pairs may have ambiguous results of ErrP decoding, with similar numbers of true and false results. And taking these pairs for ErrP labeling can decrease the uncertainty, making the RL agent more decisive on action selection. Confidence level of Binomial random variable is used here as the metric of uncertainty. Then acquisition function based on confidence interval can be expressed as

$$\sum_{t=1}^{T} \frac{1.96}{N_t^s + N_t^f} \sqrt{\frac{N_t^s N_t^f}{N_t^s + N_t^f}}$$

  where $N_t^s$ ($N_t^f$) are number of True (False) ErrP decoding results for state-action pair at step $t$ on the trajectory.

## 7.4 REWARD LEARNING

Since implicit human feedback via ErrP is noisy (hence *imperfect demonstrations*), we model the reward learning as a probabilistic maximum entropy RL problem. Following the principle of maximum entropy, given Q function $Q(\cdot, \cdot)$, the policy distribution and value function in terms of Q function can be expressed as follows,

$$V_Q(\boldsymbol{s}) = \alpha \log \sum_a \exp(Q(\boldsymbol{s}, \boldsymbol{a})/\alpha), \qquad \pi_Q(\boldsymbol{a}|\boldsymbol{s}) = \exp((Q(\boldsymbol{s}, \boldsymbol{a}) - V_Q(\boldsymbol{s}))/\alpha) \qquad (3)$$

where $\alpha$ is a free parameter, tuned empirically. The likelihood of positive and negative state-action pair are denoted as $\pi_Q(\boldsymbol{a}|\boldsymbol{s})$ and $1 - \pi_Q(\boldsymbol{a}|\boldsymbol{s})$. When demonstrations and corresponding implicit human feedback are ready, we train the Q function by maximizing the likelihood of both positive and negative state-action pairs in the demonstrations.

In order to refine the reward shape and attenuate the variance of learning updates, we introduce another baseline function $t(\boldsymbol{s})$ in the Q function. Hence, the Q function becomes $Q_B(\boldsymbol{s}, \boldsymbol{a}) := Q(\boldsymbol{s}, \boldsymbol{a}) - t(\boldsymbol{s})$. It can be proved that $Q_B(\cdot, \cdot)$ and $Q(\cdot, \cdot)$ induce the same optimal policy (Ng et al., 1999). The baseline function $t^*(\cdot)$ can be learned by optimizing $t^* = \arg\min_t J(t)$, and the objective is defined as

$$J(t) := \arg\min_t \sum_{(\boldsymbol{s},\boldsymbol{a},\boldsymbol{s}')\in\mathcal{D}\cup\mathcal{D}_R} l(Q(\boldsymbol{s}, \boldsymbol{a}) - t(\boldsymbol{s}) - \gamma \max_{\boldsymbol{a}'\in\mathcal{A}}(Q(\boldsymbol{s}', \boldsymbol{a}') - t(\boldsymbol{s}')))$$

where the loss function $l(\cdot)$ is chosen to be $l_1$-norm through empirical evaluations. In addition to the demonstration $\mathcal{D}$, we incorporate another set of demonstrations $\mathcal{D}_R$, containing transitions randomly sampled from environment without reward information. The set $\mathcal{D}_R$ is to help the function $t(\cdot)$ to efficiently learn the state dynamics, and does not require human labeling, essentially keeping the number of queries same. After reward learning, consisting of learning Q function and baseline function, for any transition tuple $(\boldsymbol{s}, \boldsymbol{a}, \boldsymbol{s}')$, the learned reward function can be represented as $Q_B(\boldsymbol{s}, \boldsymbol{a}) - \gamma \max_{\boldsymbol{a}' \in \mathcal{A}} Q_B(\boldsymbol{s}', \boldsymbol{a}')$. We then use this reward function to augment the following RL agent.

### 7.5 BOX WORLD GAME ENVIRONMENT

This environment consists of an $8 \times 8$ pixels room with keys and boxes randomly scattered. The room also contains an agent, represented by a single black pixel, which can move in four directions: up, down, left, and right. Keys are represented by a single colored pixel, and boxes are represented by two adjacent colored pixels, where the pixel on the right represents the box's lock. A key can open a lock if its color matches the lock. Its screen shot is shown in Figure 8 (a).

#### 7.5.1 EVALUATION WITH SYNTHETIC HUMAN FEEDBACK

Here, we evaluate the first framework on the Box World game (Zambaldi et al., 2018) with synthetic human feedback. This environment is introduced with details in Appendix 7.5. The synthetic feedback gives noisy label on each state-action pair, where the correct (optimal) one is labeled as wrong (sub-optimal) with the probability of $\epsilon_1$, and the wrong (sub-optimal) one is labeled as correct with the probability of $\epsilon_2$. Here, the Q network is modeled by bootstrapped DQN.

This game has a combinatorially complex environment which cannot be quickly solved by a regular RL algorithm. The simulation results are shown in Figure 8, where "No Feedback" refers to the RL algorithms without the help of human feedback. The detailed statistics of evaluations are illustrated in Table 7.5.1. Three acquisition functions are evaluated for comparison. The mean complete episode for mutual information, entropy and confidence level are $167.0, 177.5$ and $231.2$ for Human 1, $166.3, 184.2$ and $195.2$ for Human 2. We can see the first framework can achieve similar performance as full access method in terms of convergence speed, with much smaller number of inquiries. The acquisition function of confidence interval performs worst, because it does not consider the properties of the trained model. The mutual information performs better than entropy, but needs a larger number of human feedback inquiries.

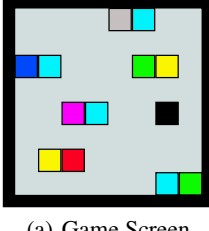

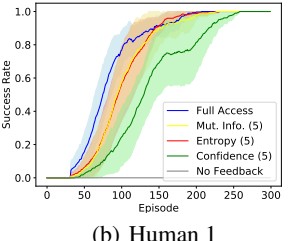

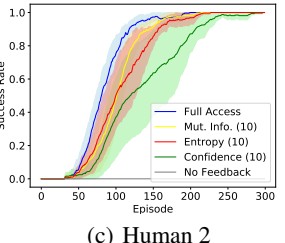

(a) Game Screen       (b) Human 1       (c) Human 2

Figure 8: First Framework Evaluation on Box World Game. The number in brackets denotes the inquiry interval $N_E$.

Table 1: First Framework Evaluation on Box World Game with Synthesized Human ($N_E$: Inquiry Interval $N_C$: Complete Episode $N_I$: Number of Inquiries). Standard deviation is presented in the brackets.

| Human | $N_E$ | Full Access $N_C$ | Full Access $N_I$ | Mutual Information $N_C$ | Mutual Information $N_I$ | Entropy $N_C$ | Entropy $N_I$ | Confidence Interval $N_C$ | Confidence Interval $N_I$ |
|---|---|---|---|---|---|---|---|---|---|
| $\epsilon_1 = 0.3, \epsilon_2 = 0.25$ | 5 | 166.9(36.2) | 7219.6(1236.0) | 167.0(36.1) | 1010.3(208.2) | 177.3(31.8) | 758.9(298.4) | 231.2(26.2) | 1003.2(154.62) |
| $\epsilon_1 = 0.2, \epsilon_2 = 0.15$ | 10 | 153.2(29.7) | 4562.2(325.3) | 166.3(33.0) | 743.3(117.5) | 184.2(61.9) | 510.6(165.3) | 195.2(30.0) | 809.9(154.6) |

Table 2: First Framework Evaluation on Maze Game with Real ErrP

| Subject | $N_E$ | Full Access | | Mutual Information | | Entropy | | Confidence Level | |
|---------|-------|-------------|---|-------------------|---|---------|---|-------------------|---|
| | | $N_C$ | $N_I$ | $N_C$ | $N_I$ | $N_C$ | $N_I$ | $N_C$ | $N_I$ |
| Subject 04 | 5 | 153.4($\pm$27.8) | 1975.4($\pm$346.2) | 161.2($\pm$21.7) | 636.7($\pm$170.3) | 172.0($\pm$47.8) | 389.3($\pm$149.9) | 168.3($\pm$22.6) | 847.2($\pm$135.2) |
| Subject 05 | 5 | 150.0($\pm$20.4) | 2130.1($\pm$357.8) | 153.1($\pm$36.8) | 505.7($\pm$221.6) | 164.8($\pm$25.4) | 394.7($\pm$109.1) | 173.8($\pm$39.6) | 887.1($\pm$292.0) |

