# OpenReview forum: "Deep Reinforcement Learning with Implicit Human Feedback"
_ICLR.cc/2020/Conference — Reject_

### Official Review · AnonReviewer3 · 2019-10-17
**Official Blind Review #3**

**Rating:** 3

**Review:**

This paper introduces several methods for training reinforcement learning agents from implicit human feedback gained through the use of EEG sensors affixed to human subjects. The EEG data is interpreted into error-related event potential which are then incorporated as a form of noisy feedback for training three different reinforcement learning agents. The first agent (full-access) requires feedback for each state-action selected. The second agent (active-learning) requires feedback on trajectories generated every N_E episodes. The third (learning from imperfect demonstrations) requires the human to provide EEG feedback over an initial set of demonstration trajectories, and subsequently requires no further human access. These methods are evaluated across several handmade gridworld-like environments including a 10x10 maze, a game of catch, and 1-D cursor game called Wobble. Using the EEG training procedures is shown to improve the speed of reaching a good RL policy for all of the three different training algorithms. Additional experiments are conducted to test the generalizability of the error-related event potentials across games, with results indicating a reasonable degree of generalization.

I like the idea of using EEG a way to reduce the burden of collecting human feedback for training reinforcement learning agents. This paper does a good job of investigating several different methodologies for combining ErrP feedback into the main loop of DQN-based RL agents. Additionally, the fact that ErrP feedback seems to generalize between domains is a promising indicator that a single person may be able to provide feedback across many different domains without re-training the ErrP decoder. While I like the paper as an interesting idea and proof of concept, there are some flaws that make me doubt it would be realizable for more complex tasks.

The drawback of this paper are the many open questions relating to the experiments:

1) In Figures 4b, 6b, and 6d, what is the meaning of 'Complete Episode'?

2) In order to assess how efficient each of these methods was in terms of the number of human labels required, how many human responses were needed for the "full-access" and "First Framework" experiments?

3) In Figure 6 - what happened to the "No Errp" baseline?

4) In Figure 5c - what are Game 1 and Game 2?

5) Why are all the results shown on the Maze domain? Why are no results shown for Catch or Wobble?

6) At an action speed of 1.5 seconds per action, I imagine that EEG is not much faster than having a human subject press a button to indicate their label. What prevents the use of faster speeds?

More broadly, I think it would be interesting to compare how effective is EEG at collecting human preferences versus pressing buttons (such as in Knox et al) or selecting preferences between trajectories (as in Christiano et al)?

It's my feeling that the experiments are more of a proof of concept and many open questions exist about whether this method would scale beyond these simple domains that DQN masters in ~300 episodes. In particular, scaling up to actual Atari games as a would go a long way towards showing scalability to a well-studied RL domain.

I thought the overall clarity of the writing was somewhat lacking with many grammatical mistakes throughout, and the necessity to refer repeatedly to the Appendices in order to understand the basic functioning of the RL algorithms and reward learning (7.4). It took several passes to understand the full approach.

**Experience Assessment:**

I have published one or two papers in this area.

**Review Assessment: Checking Correctness Of Derivations And Theory:**

N/A

**Review Assessment: Checking Correctness Of Experiments:**

I carefully checked the experiments.

**Review Assessment: Thoroughness In Paper Reading:**

I read the paper thoroughly.

---

### Official Review · AnonReviewer1 · 2019-10-23
**Official Blind Review #1**

**Rating:** 1

**Review:**

The authors propose using implicit human feedback, by means of error potential (ErrP) measured using an EEG cap, as an additional input in deep reinforcement learning. The authors first argue that even though feedback is implicit, it is still costly. To overcome this, they propose: (i) using active learning, i.e., an acquisition function, during training in order to reduce the need of ErrP's and (ii) building a model of ErrP that they fit during training and then use to simulate ErrP during test. The authors evaluate their ideas in a real experiment which consists of game playing with three different games and argue that ErrP generalize across games.

The use of ErrP in RL appears novel, however, there are several concerns that prevent me from recommending acceptance:

1. The motivation for the use of ErrP is very weak. The authors claim that it is better to use ErrP than explicit feedback by users. However, to use ErrP, the humans have to still pay attention and follow the game. In the BCI literature, ErrP have been used mainly in the context of people with disabilities who are unable to provide explicit feedback.

2. The authors use known methods to detect ErrP. As a result, basically, their system just receives a binary signal which indicates the presence of ErrP and the methodological contribution in this paper is minimal.

3. Given the minimal methodological contribution, the experiments should be much more thorough.

4. The authors only provide a hand wavy visual validation of the consistency of the potentials.

5. In section 5.2.1, the authors talk about AUC in the first paragraph, however, they do not clearly specify what was the prediction/classification task. It is also unclear overall what is the "success rate". Mor generally, the authors do not provide sufficient details of their experimental setup and evaluation metrics.

Minor comment: Last paragraph of section 2, adpoted -> adopted.

**Experience Assessment:**

I have published one or two papers in this area.

**Review Assessment: Checking Correctness Of Derivations And Theory:**

I assessed the sensibility of the derivations and theory.

**Review Assessment: Checking Correctness Of Experiments:**

I assessed the sensibility of the experiments.

**Review Assessment: Thoroughness In Paper Reading:**

I read the paper at least twice and used my best judgement in assessing the paper.

---

### Official Review · AnonReviewer2 · 2019-10-26
**Official Blind Review #2**

**Rating:** 3

**Review:**

This paper tackles the problem of obtaining feedback from humans for training RL agents, in a way that does not require too much time or mental burden. The proposed approach does this by using implicit human feedback, obtained by measuring EEG signals from the brains of human observers. When humans spot an error (e.g., in the behaviour of the RL agent), an error potential (ErrP) is generated, which differs from human to human. The proposed approach first uses supervised learning to learn what the ErrP looks like for a particular person, and then uses this to provide an additional reward signal to the agent -- the agent receives a negative reward for state-action pairs that result in an ErrP.

The goal of this paper is to show that using implicit human feedback via ErrPs is a feasible way to speed up the training of RL agents. Towards this, the paper makes two main contributions: (1) experimental evidence that the ErrP for a human can be learned for one game, and transferred zero-shot to other games, and (2) two approaches of collecting this implicit feedback from humans for a smaller number of trajectories, instead of collecting it for all trajectories encountered by the agent during learning. The experimental evaluations, on three simple Atari-like domains, show that agents learn more quickly with implicit feedback, even when gathered for a small number of trajectories, compared to not having any feedback.

This paper is well-motivated, and tackles an important problem. However, I have several concerns related to feasibility and reproducibility, as described below. The writing also needs quite a bit of editing; there are typos, grammatical errors, and incorrect / missing figure and section references.

Regarding feasibility, my first concern is that the domains considered here are relatively simple. Is there evidence that learned ErrPs would also generalize between tasks with more complex visuals? Also, in order to obtain these error signals, agent trajectories need to be slowed down significantly (from multiple actions a second to one action every 1.5 seconds). I imagine this would quickly become very tedious for people. This seems to be an inherent limitation of this approach, because relevant EEG signals happen up to 1+ seconds after an event occurs. I have doubts that this would generalize to more complex, long-horizon tasks, in which trajectories are at least hundreds of timesteps long. Finally, the study was conducted on a small number of participants (five), with a narrow range of ages and limited gender diversity. I sympathize with the difficulty in obtaining subjects for such studies, but the small sample size makes it questionable whether these results on transferability of learned ErrPs would apply to the wider population.

Reproducibility questions:
- How are the labels (of error vs. not error) obtained for learning per-person error potentials? For more complex domains, it seems that it would be harder to obtain these ground-truth labels, because if there are substantially different strategies for playing the game, humans could disagree on which state-action pairs they consider to be errors.
- In Section 4.2, how exactly does the RL agent use the saved state-action error labels (from queries to the human) to determine if the current state-action pair is an error? Is it using some form of nearest-neighbor? The notion of buckets isn't clearly described.

Additional questions and comments:
- Why does Wobble have the worst AUC for learning ErrPs (Figure 5a)? It seems to be the game that has the most obvious errors (i.e., agent moves away from the target), so I would expect it to have the best AUC.
- In Section 5.2.1 (Generalizability), what are hypotheses for *why* the ErrP trained on Wobble generalizes better to certain situations in Catch and worse for others? It would be useful to be able to characterise / predict how good generalization will be from one type of game / situation to another.
- I would like to see reports of wall-clock time for collection of implicit human feedback, for full access and the two other approaches. In other words, how long do humans have to spend connected to the electrodes?
- I'm curious how imperfect the demonstrations need to be, in order for the ErrPs for those demonstrations to provide a useful reward signal to the agent. In the paper, incorrect actions appear with a probability of 0.2, but there's no explanation for how this number was selected, and whether others were tested.

Minor comments:
- The diagram in Figure 2a is confusing because DQN is there twice —- explicitly as a component, and implicitly as part of the RL agent.
- The x-axis labels in Figure 5b don't agree with the subject labels in Figures 4b and 7b.
- In Figure 6, data for only two of the five participants is reported; it would be useful to include the results for the other participants in the Appendix.

**Experience Assessment:**

I have read many papers in this area.

**Review Assessment: Checking Correctness Of Derivations And Theory:**

I carefully checked the derivations and theory.

**Review Assessment: Checking Correctness Of Experiments:**

I carefully checked the experiments.

**Review Assessment: Thoroughness In Paper Reading:**

I read the paper at least twice and used my best judgement in assessing the paper.

---

### Author Response · Authors · 2019-11-15
**Thank you for the feedback and suggestions.**

We thank the reviewers for their valuable feedback. The concerns pointed out by the reviewers are helpful to strengthen our contributions. Since the major issues with the current version demand conducting more experiments, we have decided not to provide a rebuttal response and would be working to incorporate the feedback by reviewers into our work for the future iterations of this work.

---

### Decision · Program_Chairs · 2019-12-19

**Decision:**

Reject

**Comment:**

The paper explores the idea of using implicit human feedback, gathered via EEG, to assist deep reinforcement learning. This is an interesting and at least somewhat novel idea. However, it is not clear that there is a good argument why it should work, or at least work well. The experiments carried are more exploratory than anything else, and it is not clear that much can be learned from the results. It's a proof of concept more than anything else, of the type that would work well for a workshop paper. More systematic empirical work would be needed for a good conference paper.

The authors did not provide a rebuttal to reviewers, but rather agreed with their comments and that the paper needs more work. In light of this, the paper should be rejected and we wish the authors best of luck with a new version of the paper.